# DNA Methylation Patterns in the Early Human Embryo and the Epigenetic/Imprinting Problems: A Plea for a More Careful Approach to Human Assisted Reproductive Technology (ART)

**DOI:** 10.3390/ijms20061342

**Published:** 2019-03-17

**Authors:** Yves Menezo, Patrice Clément, Brian Dale

**Affiliations:** 1Laboratoire Clément, Avenue d‘Eylau, 75016 Paris, France; patriceclement@me.com; 2London Fertility Associates, Harley St, London W1G 7JD, UK; 3Centre for Assisted Fertilization, 80123 Naples, Italy; brian.dale@virgilio.it

**Keywords:** IVF, methylation, imprinting, epigenesis, maternal to zygotic transition, culture conditions

## Abstract

An increasing number of publications indicate that babies born after IVF (in vitro fertilization) procedures have higher rates of anomalies related to imprinting/epigenetic changes, which may be attributed to suboptimal culture conditions. Appropriate maintenance of DNA methylation during the first few days of an in vitro culture requires a supply of methyl donors, which are lacking in current in vitro culture systems. The absence of protection against oxidative stress in the culture increases the risks for errors in methylation. A decrease in the methylation processes is sometimes observed immediately post fertilization, due to delays that occur during the maternal–zygotic transition period. Care should be exercised in ART (assisted reproductive technology) procedures in order to avoid the risk of generating errors in methylation during the in vitro culture period immediately post fertilization, which has an impact on imprinting/epigenetics. Formulation of IVF culture media needs to be re-assessed in the perspective of current knowledge regarding embryo physiology.

A number of publications now indicate that babies born as a result of IVF procedures show different patterns of DNA methylation compared to babies who were conceived naturally [1,2,3]. In the majority of cases this is not due to intrinsic characteristics of the gametes, but instead to sub-optimal culture condition in vitro.

Commercial IVF media does not contain methyl donors, such as folates [4]. Methionine, the precursor for the universal methylation cofactor S-Adenosyl Methionine (SAM), is also absent in some products. This omission apparently resulted from a previous erroneous concept suggesting that some essential amino acids are toxic to early preimplantation embryos. Human oocytes express high levels of folate receptor 1 and folate transporter1 (SLC19A1), indicating that these molecules play an important role during the first 3 to 4 days of development, up to the onset of genomic activation (also known as the maternal to zygotic transition, MZT). In addition, IVF culture media spontaneously generates free radicals during incubation and has no protection against oxidative stress [5]. This may lead to oxidation of methylcytosine (MeC), causing active de-methylation of some CpG sites [6]. A pathway involving Tet3/TDG (Ten-eleven translocation/Thymine DNA glycosylase)-mediated MeC oxidation followed by loss of 5hMC (5 hydroxymethylcytosine), as a result of base excision repair, is present and active during preimplantation development, but a process that relies on passive DNA replication appears to be more active. The relative importance of the 2 processes is a controversial issue [7,8]. De-methylation and maintenance of DNA methylation both co-exist during the preimplantation stages of development and oxidative stress may create an imbalance between the ratio of the two processes.

In general, there is a direct link between oxidative stress and errors of methylation [6]. Commercial culture media has been shown to lead to imprinting defects in mouse embryos [9]. In early human embryos, metabolism up to day 3/4 is dependent upon stored maternal mRNAs and proteins deposited during oocyte growth within ovarian follicles. Storage of mRNA decreases, in both quality and quantity, with increasing age, with an effect upon most, if not all, of the metabolic pathways that are important during early development, including resistance to oxidative stress as well as methylation processes. This has an impact upon DNA stability and DNA repair processes [10].

One of the (PolyA) mRNAs with high levels of expression in the early human embryo codes is the enzyme DNMT1 (DNA Methyltransferase), which is responsible for methylation maintenance [11]. It is expressed at close to 900 times the background level, a rate that is similar to that of tubulin, one of the most common cellular structural components. The ratio of DNMT1/DNMT3A expression is roughly 7.2. DNMT3b, specifically expressed in totipotent embryonic cells, has a DNMT1/DNMT3B ratio of 6.8. In addition, the overall machinery necessary for methylation (Methionine uptake, SAM synthase, and SAH hydrolase) is expressed and active in the oocyte and the early embryo before the onset of MZT [12].

A recent paper by Smith et al., published in the journal Nature [13], suggests that the human embryo undergoes a rapid drop in methylation shortly after fertilization. DNA methylation patterns form the molecular basis for imprinting in gametes and early embryos and understanding these patterns is crucial, since alterations may lead to transgenerational epigenetic disorders, such as autism [14]. We propose that hypomethylation observed in human IVF embryos may be an artifact and side effect of poor culture conditions. Under natural conditions, methylation maintenance may attenuate any post-fertilization drop. In further support of their findings, Smith et al. [13] maintain that human and mouse DNA methylation patterns are similar, with hypomethylation occurring in both. In contrast, the decrease in methylation in mouse embryos has been shown to be gradual, with a characteristic plateau 2–3 days after fertilization [15]. This gradual decrease in mouse embryo methylation has recently been confirmed, with a relatively high quantity of methylcytosine (MeC) and only a slight decrease in the activity of the DNMTs observed at these stages [16]. In human embryos, methylation maintenance is initially high and decreases only after the 4-cell stage [17]. This means that ensuring correct methylation maintenance in the human embryo during the first 3–4 days in vitro requires adequate support/supplementation, before the drop in methylation occurs. DNA methylation also has a profound impact on genome stability. Recent observations obtained from in vitro fertilization (IVF) and preimplantation genetic screening (PGS) treatment cycles have provided confirmation that women carrying the C677T MTHFR SNP (methylenetetrahydrofolate reductase single nucleotide polymorphism) generate preimplantation embryos with high rates of aneuploidy and a dramatic decrease in viability. [18]. This SNP and, to a lesser extent, the A1298C MTHFR SNP are known to impair the supply of folate, especially in individuals who are homozygous for the SNP. This may also explain the efficiency of “in vivo” treatment with 5MTHF (5-methyltetrahydrofolate) supplements before and during pregnancy. These SNP carriers suffer long-lasting infertility and repeated miscarriages [19]. 5MTHF is the folate compound located immediately downstream of MTHFR and, therefore, can by-pass the problem that is caused by MTHFR SNPs.

These major biochemical pathways must be taken into account, with respect to formulation of embryo culture media, in order to avoid errors in methylation/epigenetics immediately post-fertilization and also imprinting errors in particular, as recently described in the literature [20]. Finally, the use of embryonic stem cells as a model for early human embryo metabolism, as described by Smith et al. [13] is not appropriate. There is no alternative existing biological model for mammalian preimplantation embryo development in the period prior to activation of the zygotic genome (MZT).

Until methylation patterns in the human embryo can be measured in vivo, or conditions for in vitro culture are known to be adequate, data generated from embryos created by IVF must be interpreted with caution, in particular with reference to the knowledge that IVF babies may have altered patterns of DNA methylation.

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
