# Peer review of "DNA Methylation Patterns in the Early Human Embryo and the Epigenetic/Imprinting Problems: A Plea for a More Careful Approach to Human Assisted Reproductive Technology (ART)"

_ijms, 2019, doi:10.3390/ijms20061342_

Round 1
Reviewer 1 Report
In the commentary manuscript by Yves Menezo et al., authors discussed imprinting/epigenetics anomalies on the babies born from In-Vitro Fertilization (IVF) procedures, providing a critical viewpoint on careful application of human assisted reproductive technology (ART). The authors precisely clarified viewpoints, however, the writing of this manuscript needs a further polish. Therefore, I would suggest this manuscript should undergo a textual improvement for future publication in International Journal of Molecular Sciences according to points below:
1. There are written and grammar errors and authors should read carefully and revise. For example, page 1 line 26, “methylation/Imprinting/epigenetics errors”.
2. Abbreviation should be defined in the text at first use.
3. The references should be presented in a style consistent with the journal guideline.
Author Response
The paper has been fully re modified in "Cambridge" English by Dr Kay Elder, pH D, MD scientific consultant at the Bourn Hall Clinic
Abbreviation have been defined in the text
The references have been modified according to MDPI requests and doi added for all the papers
Reviewer 2 Report
In the manuscript “DNA methylation patterns in the early human embryo and the Epigenetic/imprinting problems A plea for a more careful approach to Human ART”, the authors pointed out an issue that the maintenance of methylation in embryos is crucial in IVF procedures. They overviewed by comparing the level of DNA methylation in the embryos that were from in vitro culture or from in vivo. They further argued that the rapid drop of DNA methylation in the embryos that were from in vitro culture was due to the culture condition that lack of protection against oxidative stress.
Overall, the commentary presented is interesting and clear. However, several concerns with this manuscript that needs to be further clarified to make the issue comprehensive.
1. When they were discussing the negative impact of oxidative stress in the maintenance of DNA methylation in the IVT process, DNA de-methyl-transferases need to take into account this de-methylation in order to make the issue comprehensively. The authors should make more words with additional references such as Inoue A. et al., 2011, Science; Shen L. et al., 2013, Cell Stem Cell and Wang L et al., 2014, Cell and so on.
2. InLine 55, the underscore by the words of “ totipotent embryonic cells” should be removed.
Author Response
As requested we have added the refs of Inoue (with doi) and Wang. et al. (with doi) They are rather complementary for active vs passive demethylation in relation with oxidative stress.
7 lines of explaination have been added
Underlined removed (Stem cells)
Reviewer 3 Report
In this commentary, Menezo and colleagues have briefly stated DNA methylation issues in the early human embryo and the relevant epigenetic problems, particularly in the embryo culture media during the IVF-ET procedures. Overall, it is concise and clear. However, there are several specific points needing to be clarified.
1. after reading the reference 1-3, paragraph 1, it needs to be cautious to note that there are some differences in the baseline characteristics, such as age, in the literature, which is known tto be an important factor for DNA methylation.
2. Some culture IVF media from Vitrolife is supplemented with human serum which may contain folates.
3. Would be better to add more explanations on the de-methylation process during embryo culture in vitro.
4. Administration of Gn and other factors need to be taken into consideration as well when discussing methylation of IVF embryos.
5. At the first time of introducing an abbreviation, it would be better to give the full name, for example: SAM.
Author Response
We have added the effect of age on all the biochemical parameters of the embryo. THe paper of Hamatani has been added in refs. (quoted as reference 10)
We have also added the negative impact of MTHFR SNPs : reference of Enciso added as ref 18
Decreased capacity to sustain the 1-CC and then methylation, affect embryo quality
For Culture media: It is usually either forbidden/not recommended to add serum in IVF culture media, especially if the infertility is idiopatic. Some media have SSR as complement: Serum substitute : dialysed and fractionned serum proteins , not intermediate metabolism molecules so no methyl donors
Demethylation: we have added the refs 7 and 8! Inoue and Zhang , and wang et al. 7 lines of explainations have been added . It is also well explained in the paper with ref 6
Little have very few papers concerning Gn and methylation. Moreover the paper ref 9 (mouse) and 18 (human) are not in favor of this concept even if it cannot be excluded in PCOS, hyperstimulation
We have added the definitions for all the abbreviations written in text
Round 2
Reviewer 3 Report
The commentary is much better after revision.